# Assessment of Food Hygiene Non-Compliance and Control Measures: A Three-Year Inspection Analysis in a Local Health Authority in Southern Italy

**DOI:** 10.3390/foods14193364

**Published:** 2025-09-28

**Authors:** Caterina Elisabetta Rizzo, Roberto Venuto, Giovanni Genovese, Raffaele Squeri, Cristina Genovese

**Affiliations:** 1Department of Prevention, Local Health Authority of Messina, 98123 Messina, Italy; caterina.rizzo93@gmail.com (C.E.R.); roberto.venuto@hotmail.it (R.V.); 2Department of Biomedical and Dental Sciences and Morphofunctional Imaging, University of Messina, 98124 Messina, Italy; squeri@unime.it; 3Department of Chemical, Biological, Pharmaceutical and Environmental Sciences, University of Messina, 98166 Messina, Italy

**Keywords:** food safety, HACCP, public health, RASFF

## Abstract

Background and Aim: Food hygiene is fundamental to public health, ensuring safe and nutritious food free from contaminants, and is vital for economic development and sustainability. The Hazard Analysis and Critical Control Points (HACCP) system is a crucial tool for managing risks in food production. Despite global recognition of food safety’s importance, significant disparities exist, especially in Southern Italy, where diverse food production, tourism, and economic factors pose challenges to enforcing hygiene standards. This study evaluates non-compliance with food hygiene regulations within a Local Health Authority (LHA) in Calabria, Southern Italy, to inform effective public health strategies. Materials and Methods Authorized by the Food Hygiene and Nutrition Service (FHNS) of the LHA, the study covers January 2022 to December 2024, analyzing 579 enterprises with 1469 production activities. Inspections followed EC Regulation No. 852/2004, verifying the correct application of procedures based on the Hazard Analysis and Critical Control Points (HACCP) principles, including the operator’s monitoring of Critical Control Points (CCPs), and adherence to Good Hygiene Practices (GHPs). Non-compliances were classified by severity, and corrective and punitive actions were applied. Data were analyzed annually and across the full period using descriptive statistics and chi-squared tests to assess trends. Results: Inspection coverage increased markedly from 29.8% of production activities in 2022 to 62.5% in 2023, sustaining 62.0% in early 2024, exceeding the growth of new activities. Inspections were mainly triggered by RASFF alerts (22.4%), routine controls (20.0%), and verification of previous prescriptions (14.3%). The most frequent corrective measures were long-term prescriptions (28.6%), violation reports (22.9%), and short-term prescriptions (20.0%). Enterprises averaged 4.61 production activities, highlighting operational complexity. Conclusions: This study provides a granular analysis of food hygiene non-compliance within a Local Health Authority (LHA) in Southern Italy, to inform effective public health strategies. While official control data may be publicly available in some contexts, our research offers a unique, in-depth view of inspection triggers, non-compliance patterns, and corrective measures, which is crucial for understanding specific regional challenges. The analysis reveals that the prevalence of long-term prescriptions and reliance on RASFF alerts indicate systemic challenges requiring sustained interventions.

## 1. Introduction

Food hygiene constitutes a fundamental pillar of public health protection by ensuring that food consumed by individuals is safe, nutritious, and devoid of harmful contaminants [1]. Furthermore, food safety has emerged as a global priority, integral to economic development and sustainable growth [2]. A comprehensive approach to food safety transcends the mere prevention of foodborne illnesses; it necessitates the coordination of multiple stakeholders throughout the entire food chain, from primary production to the end consumer, underpinned by a thorough understanding of production, distribution, and consumption dynamics [3].

Within this framework, the Hazard Analysis and Critical Control Points (HACCP) system represents a pivotal risk management tool. Its implementation facilitates proactive identification and control of potential hazards through systematic analysis of all production phases and the application of targeted interventions at critical control points [4]. It is important to note that Regulation (EC) No. 852/2004 requires food business operators to implement and maintain procedures based on the HACCP principles. However, it also allows for flexibility, particularly for small businesses. In many cases, adherence to officially recognized Guides to Good Hygiene Practice, which are encouraged by the Regulation, is considered sufficient to meet the legal requirements for managing food safety risks, without the need for a formal, complex HACCP system. Food safety, therefore, encompasses a “farm to fork” continuum, extending beyond the prevention of physical, chemical, and microbiological contamination to include aspects such as accurate labeling, product traceability, allergen management, and nutritional considerations [5,6].

The accelerating processes of globalization and international trade underscore the necessity for harmonized food safety standards globally, fostering uniform practices and regulatory coherence [2]. Despite widespread acknowledgment of food safety’s paramount importance, significant disparities persist internationally regarding the adoption and enforcement of safety standards [7,8].

Within the European Union (EU), stringent regulatory frameworks establish comprehensive hygiene standards applicable across the food supply chain, enforced primarily through official controls conducted by national and local health authorities [9,10]. These mechanisms aim to ensure compliance by food business operators at every stage—from production to retail—thereby mitigating risks of foodborne diseases and safeguarding consumer health.

This entire framework is built upon the general principles and requirements laid down by Regulation (EC) No. 178/2002, often referred to as the ‘General Food Law.’ This foundational regulation established the European Food Safety Authority (EFSA) and defined key concepts such as risk analysis, traceability, and the primary responsibility of food business operators for ensuring food safety.

In Italy, food safety legislation is grounded in both EU directives and national laws, with enforcement responsibilities delegated to local health authorities [11]. These EU regulations are implemented and complemented by a framework of national laws and regional decrees that define the specific operational procedures for official controls and sanctions, adapting the broader European principles to the Italian context. These bodies are charged with inspecting food establishments, monitoring hygiene practices, and ensuring adherence to applicable regulations. Nonetheless, consistent compliance remains a challenge, particularly in certain regions. Non-compliance poses significant risks, including contamination events, disease outbreaks, and diminished consumer confidence in food safety systems [12]. Table 1 outlines a proportional and proactive approach to food safety management. It highlights how the intensity of the response must match the severity of the risk, emphasizing the importance of not only fixing existing problems but also preventing future ones by addressing their root causes.

While the European Union has established a robust regulatory framework, consistent compliance remains a significant challenge, particularly in certain regions. The effectiveness of official controls in Southern Italy has been questioned, with potential contributing factors including limited awareness of hygiene regulations, resource constraints, cultural perceptions, and economic pressures. This study aims to move beyond simple reporting of compliance rates by providing an in-depth, three-year analysis of inspection data to understand the underlying nature of non-compliance in a local context.

Southern Italy exemplifies a region confronted with unique difficulties in implementing food hygiene standards due to its heterogeneous factors that can be hypothesized to contribute to non-compliance in several ways:Tourism-driven demand: The intense seasonal pressure on food businesses can lead to overburdened facilities, the hiring of temporary and less-trained staff, and shortcuts in hygiene procedures to meet high demand.Heterogeneous food production: The prevalence of small, family-run businesses often utilizing traditional methods may coexist with a lack of resources for structural investments, leading to infrastructural deficiencies that require long-term corrective actions.Socioeconomic disparities: Economic pressures may disincentivize investments in modern equipment, staff training, and robust self-control systems, making compliance a secondary priority for some operators.

The effectiveness of official controls in this region has been questioned, with persistent non-compliance across various local enterprises. Contributory factors may include limited awareness of hygiene regulations, resource constraints impacting inspection capacity, cultural perceptions of food safety, and economic pressures [13].

Understanding the prevalence and underlying causes of non-compliance is critical to devising targeted, effective interventions. This study aims to evaluate the extent of food hygiene non-compliance within a Local Health Authority (LHA) in Southern Italy. By analyzing inspection data and identifying trends, the study seeks to elucidate challenges faced by local regulators in enforcing hygiene controls. Ultimately, this research endeavors to inform strategies that enhance food safety practices, promote regulatory adherence, protect public health, and reinforce consumer trust in the safety of food products within the region.

## 2. Materials and Methods

### 2.1. Setting

The study was conducted with the permission of the Food Hygiene and Nutrition Service (FHNS) (in Italian: SIAN, *Servizio Igiene degli Alimenti e Nutrizione*) of a Local Health Authority in Calabria, a region of Southern Italy, for three years, from January 2022 to December 2024. An exhaustive census of all inspections conducted during the study period has been conducted. A complete, anonymized administrative dataset of all official inspections has been provided by the LHA. The selection by the FHNS aimed to include a representative cross-section of the local food industry, which predominantly consists of small and medium-sized enterprises (SMEs). The activities analyzed spanned various sectors, including catering (restaurants, bars), retail (supermarkets, small grocery stores), and small-scale production (bakeries, pasta factories, confectioneries).

This study focused on operators falling under Regulation (EC) No. 852/2004. Establishments requiring specific approval under Regulation (EC) No. 853/2004 (dealing with food of animal origin for intra-community trade) were not included in this specific dataset, which represents a scope limitation of our analysis

The sample included businesses of different sizes and industries within the food supply chain, selected by the FHNS to represent the regional picture under consideration in a heterogeneous way. The data was analyzed using R software (version 4.4.0).

### 2.2. Inspection Procedures

The inspections were conducted by qualified LHA personnel following a standardized institutional checklist. This checklist is based on the general hygiene requirements for all food business operators as laid out in Annex II of Regulation (EC) No. 852/2004. The assessed areas systematically included premises, transport, equipment, food waste management, water supply, personal hygiene, and provisions for foodstuffs. It is important to note that this study did not analyze the specific results of laboratory sampling (e.g., detection of specific microorganisms or chemical contaminants), but rather the frequency and nature of the administrative actions resulting from on-site inspections of processes and structures.

Inspection operations included verifying the implementation of good hygiene practices (GHPs) and the correct application of procedures based on HACCP principles, which involves assessing the operator’s own monitoring of critical control points (CCPs) at various phases of production. Particularly, the assessed areas systematically included:General requirements for food premises (e.g., layout, design, construction, site, and size);Specific requirements in rooms where foodstuffs are prepared, treated, or processed (e.g., floors, walls, ceilings, windows, ventilation);Requirements for transport;Equipment requirements (e.g., materials in contact with food, cleaning procedures, maintenance);Food waste management;Water supply;Personal hygiene (e.g., training, cleanliness, and health status of food handlers);Provisions applicable to foodstuffs (e.g., temperature control, prevention of cross-contamination);Provisions applicable to the wrapping and packaging of foodstuffs;Training of personnel.

There are several reasons why LHA inspectors carried out inspections on food businesses. Table 2 provides a key to the reasons behind the food business inspections conducted by the Local Health Authority (LHA) in Southern Italy. The inspections are carried out in compliance with the general principles and requirements laid down by Regulation (EC) No. 178/2002, known as the ‘General Food Law’. This foundational regulation establishes key concepts like risk analysis and the primary responsibility of food business operators (FBOs) for ensuring food safety. The table shows various inspection triggers, from routine controls (B.05) to reactive measures like those prompted by RASFF alerts (B.02), as well as follow-ups for verification of prescriptions (B.07). These triggers highlight a dual strategy of proactive oversight and reactive intervention.

### 2.3. Types of Non-Compliances and Sanctions

Non-compliances can be classified as serious and minor, based on the severity of the risk they pose to public health and adherence to regulations. Sanctions in the food sector are a fundamental tool for ensuring compliance with food safety and hygiene regulations: they serve as both a deterrent and a corrective means of influencing the behavior of food business operators (FBOs), with the primary objective of protecting public health and maintaining high food safety standards. These can vary significantly based on the severity of the non-compliance and the quantities found during official controls.

It is important to clarify the scope of the non-compliances analyzed in this paper. Our study focuses on non-compliances related to hygiene requirements and procedures identified during official inspections, as per Regulation (EC) No. 852/2004. These relate primarily to failures in Good Hygiene Practices (GHPs) and procedural requirements based on HACCP principles.

This study did not analyze the specific results of laboratory sampling (e.g., detection of specific microorganisms like Listeria monocytogenes or Salmonella, or chemical contaminants). While such sampling is a part of the LHA’s overall control activities, our research objective was to analyze the frequency and nature of the administrative actions resulting from on-site inspections of processes and structures, not the outcomes of laboratory tests.

Table 3 highlights a graduated and proportionate approach to non-compliance: starting from “softer” corrective measures (prescriptions, extensions, follow-up) up to more severe interventions (goods destruction, suspension of activities) when the public health risk is high. This framework aligns with the principles of progressive enforcement established by European regulations, aiming to balance deterrence with support for continuous improvement and it is contained in Law Decree 27-2021.

### 2.4. Data Analysis

To assess the evolution of inspections and non-compliances found, the data was examined both annually and in the entire three-year period. The distribution of inspections among the various manufacturing units was determined using statistical tools, and notable differences in the years under investigation were noted. The information was further separated into inspection motive categories, including routine inspections (B.05) and Rapid Alert System for Food and Feed (RASFF) reports.

The chi-squared (χ2) test was used to build contingency tables to evaluate the hypotheses. Only when the null hypothesis (H0) was rejected and r × k tables were present was the method of splitting the degrees of freedom used. R software version 4.4.0 was used for the statistical studies, both synthetic and inferential.

### 2.5. Ethical Statement

Official authorization to access and use the inspection data for research purposes was obtained from the Director of the Food Hygiene and Nutrition Service (FHNS) of the Local Health Authority. The study was conducted in full compliance with data protection regulations. All data provided to the research team were fully anonymized by the LHA prior to analysis, with all identifying information related to specific enterprises or individuals removed. As this research involved the analysis of pre-existing, non-identifiable administrative data, a formal ethical review by an institutional committee was not required according to national guidelines.

## 3. Results

### 3.1. Overview of the Sample

Over a three-year period, from January 2022 to December 2024, this study analyzed inspection activities carried out by the Food Hygiene and Nutrition Service (FHNS) of a Local Health Authority (LHA) in Calabria, Southern Italy. The sample comprised 579 enterprises encompassing a total of 1469 distinct production activities. Given the classification of each enterprise according to its specific manufacturing processes, inspections revealed an average of 4.61 production activities per enterprise.

### 3.2. Inspection Trends Across the Three-Year Period

A year-by-year analysis shows a significant increase in the frequency and coverage of inspections, as illustrated in Figure 1. In 2022, the number of active production activities was 436, and with 130 inspections conducted, the inspection rate was 29.8%; in 2023, it rose to 62.5% (n = 356); and in the first seven months of 2024, it remained at 62.0% (n = 287).

### 3.3. Reasons for Inspections

The main reasons for inspections were RASFF (B.02) (22.4%), followed by official controls (B.05) (20%) and verification of prescriptions (B.07) (14.3%). All the reasons are reported in Figure 2.

This distribution underscores the LHA’s emphasis on addressing immediate risks through targeted inspections, with RASFF reports and routine controls forming the bulk of activities.

### 3.4. Measures Adopted During Inspections

During the inspections, various corrective and enforcement measures were implemented based on the severity of non-compliance: long-term prescriptions (A.04) accounted for the highest percentage at 28.6%, followed by reporting violations (A.07) (22.9%) and by short-term prescriptions (A.03) (20%). A smaller percentage of actions, such as the destruction of goods or suspension of laboratory activity, were reserved for severe cases. All the measures adopted are shown in Figure 3.

## 4. Discussion

This study offers a comprehensive real-world overview of official food hygiene inspections conducted by a Local Health Authority (LHA) in Calabria, Southern Italy, over the period 2022–2024. Our research is unique not for the availability of the data, but for the depth of analysis we provide on the distribution of inspection triggers and the nature of corrective actions. This approach allows us to draw specific lessons about the challenges faced in securing regulatory compliance in this particular region. Our finding that long-term prescriptions are the most frequent corrective measure (28.6%), suggesting a prevalence of structural deficiencies, outdated procedures, or insufficient staff training, requiring sustained intervention rather than immediate rectification. This aligns with concerns raised in other economically heterogeneous regions. In contrast, regions with more established food industries often report a higher proportion of non-compliances related to documentation or minor procedural errors. This suggests that the challenges faced in Southern Italy may be more fundamental, requiring investments in infrastructure and training rather than simple procedural adjustments.

This finding is consistent with reports from economically heterogeneous regions of Europe, where small and medium-sized enterprises (SMEs) face difficulties in allocating sufficient resources for infrastructure upgrades, continuous staff training, and robust implementation of HACCP-based systems. The reliance on RASFF notifications (22.4%) as one of the main triggers for inspections underlines the reactivity of the current system. While RASFF remains a cornerstone of European food safety governance, the heavy dependence on alerts suggests that preventive mechanisms, such as routine inspections and internal self-monitoring, may not yet be fully effective in the studied context. In contrast, countries with more consolidated food safety systems often demonstrate a higher proportion of inspections stemming from proactive oversight rather than external alerts. This discrepancy calls for a shift toward strengthening preventive inspections, which would reduce the need for emergency responses. The frequent issuance of violation reports (22.9%) further illustrates a graduated enforcement strategy, where inspectors initially promote remediation but resort to formal sanctions when non-compliances persist or present heightened risks. This tiered approach aligns with the EU principle of proportionality in enforcement but raises questions about the sustainability of corrective actions in SMEs, especially when economic pressures disincentivize compliance. Supporting enterprises through targeted financial incentives, training programs, and technical assistance may enhance compliance while reducing the recurrence of violations.

Another critical element emerging from this study is the complexity of food businesses in the region, with enterprises managing an average of 4.61 distinct production activities. Such operational diversity increases the difficulty of maintaining consistent hygiene standards across different processes. It also underscores the importance of digitalization and integrated food safety management systems, which could improve traceability, streamline HACCP documentation, and facilitate internal audits, particularly in resource-limited enterprises.

From a broader perspective, these results resonate with international literature emphasizing the role of food safety culture in ensuring compliance. Beyond structural investments, fostering an organizational mindset that prioritizes hygiene, transparency, and accountability is essential. Initiatives to embed food safety culture into SMEs—through continuous education, peer-to-peer learning networks, and sector-wide campaigns—could play a transformative role.

A marked increase in inspection coverage—from 29.8% in 2022 to over 62% in 2023 and the first seven months of 2024—reflects a significant intensification of monitoring efforts across the region. This upward trajectory likely results from both reactive responses to emerging regulatory or public health concerns and proactive strategies aimed at enhancing compliance through increased oversight. Notably, the rate of inspections outpaced the growth in new production activities, indicating deliberate resource allocation and prioritization to strengthen food safety governance.

The distribution of inspection triggers reveals a balanced approach between reactive interventions, such as Rapid Alert System for Food and Feed (RASFF) notifications (22.4%), and proactive measures, including routine controls (20.0%) [14]. This dual strategy highlights the ongoing relevance of the EU’s rapid alert system in addressing imminent risks, alongside a strong commitment to preventive oversight [15]. Additionally, the significant proportion of follow-up inspections (14.3%) to verify corrective action implementation underscores systematic efforts to close compliance gaps and prevent recurrence. The distribution of inspection triggers in this study reflects a balanced operational strategy by the LHA, combining proactive oversight with reactive intervention. Proactive actions, such as official routine controls (B.05) and joint inspections with other authorities (B.01, B.06, B.08, B.11), are essential to maintain a consistent baseline of food safety compliance across the sector. Conversely, reactive triggers—most notably RASFF notifications (B.02), NAS reports (B.04), and health constraints (B.03)—demonstrate the system’s responsiveness to urgent or high-risk situations. The considerable proportion of follow-up inspections to verify prescriptions (B.07) further underscores a commitment to closing compliance gaps and preventing recurrence of non-conformities. This combination of preventive monitoring and targeted, event-driven interventions aligns with best practices in regulatory enforcement, ensuring that resources are directed both toward early detection of potential risks and swift containment of confirmed threats.

Analysis of corrective measures further clarifies the nature of non-compliance. The predominance of long-term prescriptions strongly suggests that many deficiencies may be structural or systemic- such as inadequate infrastructure. This hypothesis, while plausible, would require further investigation with data stratified by enterprise size and type, requiring sustained intervention rather than immediate rectification [16]. The frequencies of violation reports (22.9%) and short-term prescriptions (20.0%) suggest a graduated enforcement approach, initially emphasizing remediation before escalating to sanctions when necessary. This tiered methodology aligns with established regulatory compliance models that balance deterrence with corrective guidance proportionate to risk severity and recurrence [17].

The average of 4.61 production activities per inspected enterprise reveals operational complexity that may exacerbate compliance challenges. Enterprises managing multiple production lines or diverse processes face increased difficulty maintaining consistent hygiene standards, particularly in the absence of robust HACCP implementation or effective internal controls [18]. For inspectors, this complexity complicates thorough assessment, especially in settings constrained by limited resources. Similar findings are in line with previous documented experiences in European contexts, where innovative approaches to official verification have contributed to maintaining high standards of food safety even in critical periods, such as the COVID-19 pandemic [19].

While findings correspond with broader concerns documented in the literature regarding food safety in economically heterogeneous regions of Southern Italy, caution is warranted in generalizing results beyond the specific LHA under study. Variability in inspection protocols, regional economic conditions, cultural attitudes toward food hygiene, and the availability of technical support may all significantly impact compliance outcomes [20]. However, it is important to acknowledge a potential selection bias inherent in the dataset. Since the inspection plan is risk-based, including a high proportion of follow-up inspections (14.3%) and responses to RASFF alerts (22.4%), businesses with known prior issues are likely overrepresented compared to the general FBO population. Therefore, the prevalence of non-compliance reported here may be higher than the average across all businesses in the region. This is a typical feature of official control data and should be considered when interpreting the results. This research also emphasizes the critical role of clear documentation and communication between regulators and food businesses. Deficiencies such as inadequate HACCP documentation or inconsistent record-keeping, though often classified as minor or non-immediate risks, may cumulatively undermine food safety resilience and impede traceability during contamination events. Promoting enhanced internal audits, digital record-keeping systems, and ongoing staff training represents a vital strategy to address these issues.

These findings contribute to the wider discourse on the role of local authorities within the European food safety framework. In decentralized enforcement systems such as Italy’s, where regional health services bear significant responsibility, disparities in inspection frequency, thoroughness, and follow-up can lead to uneven levels of consumer protection. Addressing these disparities requires harmonized national oversight, targeted funding, specialized inspector training, and risk-based prioritization—particularly in regions characterized by socioeconomic vulnerability or high volumes of food tourism.

Finally, this study sheds light on the regional disparities that can arise within decentralized enforcement systems such as Italy’s. While national and European regulations establish common standards, the actual effectiveness of enforcement depends on local inspection capacity, resource allocation, and socio-economic context. The observed increase in inspection coverage (from 29.8% to over 62%) reflects commendable progress; however, the persistence of structural deficiencies suggests the need for long-term, systemic interventions that go beyond inspection alone. Harmonized national oversight, coupled with targeted support to vulnerable regions, may help reduce inequalities in food safety compliance and reinforce consumer trust across the country.

While findings correspond with broader concerns documented in the literature regarding food safety in economically heterogeneous regions of Southern Italy, caution is warranted in generalizing results beyond the specific LHA under study. Variability in inspection protocols, regional economic conditions, cultural attitudes toward food hygiene, and the availability of technical support may all significantly impact compliance outcomes. Our work, therefore, serves as a crucial baseline for future comparative studies.

### Limitations

A limitation of this study is the nature of the available data, which categorized non-compliances based on the resulting administrative action rather than the specific source of the deficiency (e.g., structural, equipment, personnel hygiene). Consequently, while we can identify that long-term prescriptions were common—suggesting structural issues—we cannot provide a precise statistical breakdown of the root causes. Future research should aim to collect data using a more granular classification system to better pinpoint recurring problem areas.

## 5. Conclusions

This study highlights an increasing commitment by the examined Local Health Authority (LHA) in Southern Italy to ensure food safety, evidenced by an increase in inspections and the application of corrective measures. The prevalence of Rapid Alert System for Food and Feed (RASFF) reports as a driver for inspections and the need for long-term prescriptions are two key lessons from our analysis. Recent literature highlights how the adoption of integrated food safety management and quality control strategies can improve the effectiveness of regulatory interventions, especially in contexts characterized by production complexity [21]. This underscores the need for ongoing vigilance, targeted support for Food Business Operators (FBOs), and further research to optimize control strategies and enhance the food safety culture [22]. The analysis of food hygiene non-compliance reveals recurring critical issues that demand strengthened preventive measures to safeguard public health. Furthermore, at the European level, non-compliances of lower risk are managed through the Administrative Assistance and Cooperation Network (AAC) [23] and the EU Agri-Food Fraud Network (FFN) who also operates as a complementary system for addressing non-compliances, enhancing information exchange and coordinated responses among Member States, and thereby reinforcing the overall integrity and effectiveness of the European food safety framework [24]. The analysis of food hygiene non-compliance reveals recurring critical issues that demand strengthened preventive measures to safeguard public health [25,26,27,28,29,30,31,32,33,34]. This imperative aligns with recent findings that demonstrate persistent vulnerabilities in controlling infectious risks across various community and institutional settings.

Future research should analyze the specific nature of non-compliances in relation to FBO characteristics, compare data with other LHAs to identify broader patterns, and assess the long-term impact of sanctions and corrective measures. In this framework, the implementation of systems based on risk analysis and preventive controls (HARPC) [35] is established as a complementary and sometimes more flexible approach than HACCP, offering useful tools for the improvement of food safety practices even in productions intended for vulnerable population groups Ultimately, strengthening training for FBOs, increasing resources for inspections, and promoting a culture of food safety are crucial for ensuring greater public health protection and consumer confidence [36].

## Figures and Tables

**Figure 1 foods-14-03364-f001:**
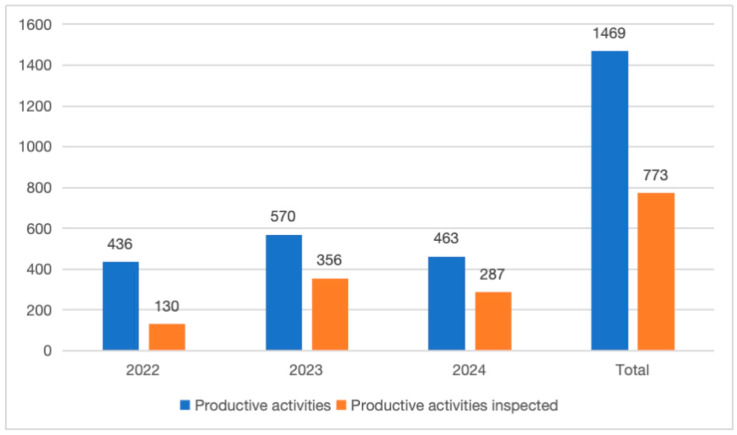
Comparison between the total number of productive activities and the number of inspected activities per year. The vertical axis represents the absolute count of productive activities, while the horizontal axis indicates the year of inspection.

**Figure 2 foods-14-03364-f002:**
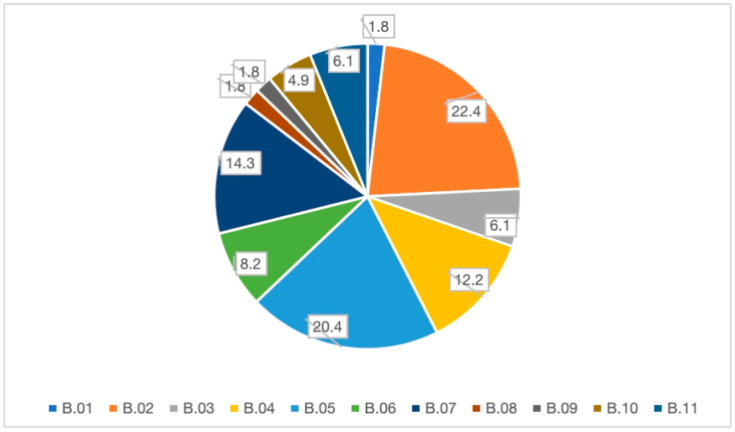
Reasons for inspections. The values represent the percentage of total inspections triggered by each specific reason. Each inspection was assigned a single primary trigger, so there are no duplicates in the data, and the sum of all categories equals 100%.

**Figure 3 foods-14-03364-f003:**
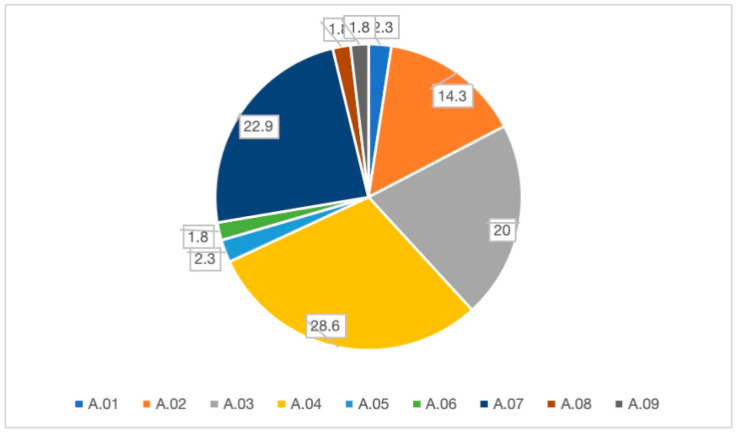
Adopted corrective measures.

**Table 1 foods-14-03364-t001:** Types of non-compliance.

Type of Non-Compliance	Risk Level	Examples	Action Required	Relevant Regulations
**Serious Non-Compliance**	Immediate risk to health	- Dangerous pathogens (Salmonella, Listeria, *E. coli*) - Chemical contamination (pesticides) - Poor hygienic conditions	Immediate suspension, product withdrawal, fines, criminal penalties	Reg. (EC) No. 852/2004 (Hygiene of foodstuffs) Reg. (EC) No. 178/2002 (General food law)
**Minor Non-Compliance**	No immediate risk to health	- Incorrect labeling - Minor HACCP documentation issues - Small temperature deviations	Corrective measures, administrative sanctions	Reg. (EC) No. 178/2002 (Food safety management) Legislative Decree No. 190/2006
**Inadequacies**	Potential future risk	- Poor maintenance of equipment - Inadequate staff training - Weak HACCP plan application	Address weaknesses, preventive measures	Reg. (EC) No. 852/2004 (Hygiene of foodstuffs) Art. 5 of Reg. 852/2004 (Food safety management system)

**Table 2 foods-14-03364-t002:** Reasons for food businesses inspections.

Code	Reason for Inspection	Description
B.01	Joint control with Border Inspection Posts (BIPs, in Italian *PIF*)	Inspections carried out in collaboration with border control facilities, typically focusing on imported food products.
B.02	RASFF	Inspections initiated following notifications from the Rapid Alert System for Food and Feed, targeting urgent food safety risks.
B.03	Health constraint	Inspections due to specific public health concerns, such as outbreaks or suspected contamination events.
B.04	Official control after the report of Anti-adulteration and Health Units of the Carabinieri police (in Italian *NAS*, *Comando Carabinieri per la Tutela della Salute*)	Inspections following alerts from the Carabinieri’s Anti-Adulteration and Health Units (NASs), usually linked to suspected fraud or serious violations.
B.05	Official routine control	Scheduled, preventive inspections carried out as part of the LHA’s regular monitoring activities.
B.06	Joint official control with various law enforcement agencies	Inspections coordinated with non-specialized police or other regulatory bodies to address cross-sector issues.
B.07	Official control for the verification of prescriptions	Follow-up inspections to ensure that previously issued corrective measures have been implemented.
B.08	Joint official control with the Service for prevention and safety in the workplace (in Italian *SPISAL* or *SPRESAL*, *Servizio per la prevenzione e la sicurezza negli ambienti di lavoro*)	Inspections conducted together with occupational health and safety authorities, often in settings where hygiene overlaps with worker safety.
B.09	Goods destruction	Inspections associated with the disposal of unsafe products, ensuring proper handling and compliance with regulations.
B.10	Official block	Inspections resulting from, or aiming to impose, a formal halt on certain operations or product movements.
B.11	Joint official control with SVET (Veterinary Service)	Inspections carried out with veterinary authorities, particularly for products of animal origin.

**Table 3 foods-14-03364-t003:** Measures taken by LHA inspectors (Law Decree 27/2021).

Code	Description	Measures Taken
A.01	A drastic measure applied in cases of immediate risk to public health (e.g., severe microbiological or chemical contamination). It indicates a critical failure in preventive measures.	Goods destruction
A.02	A positive outcome, showing that the operator has implemented the corrections required in previous inspections.	Previous prescriptions remedied
A.03	Associated with non-compliances that can be quickly resolved, such as documentary corrections or minor operational adjustments.	Short-term prescription (5–15 days)
A.04	Structural or organizational non-compliances requiring more complex interventions and investment (e.g., renovations, equipment replacement).	Long-term prescription (20–30 days)
A.05	Specific risks exist in processing or testing areas, potentially having a significant impact on production continuity.	Laboratory activities suspended
A.06	Presence of unapproved storage areas, often linked to traceability issues and potential contamination risks.	Unauthorized storage
A.07	Formal report of an infraction, with possible legal implications and financial penalties.	Reporting violation
A.08	Degree of administrative flexibility, granting the operator more time to complete corrective actions, often for technical or economic reasons.	Extension requested and granted
A.09	Obligation to officially notify the completion of corrective actions, useful for subsequent verification and formal closure of the non-compliance.	End of work reporting requirement

## Data Availability

The data analyzed in this study are not publicly available due to privacy and confidentiality restrictions imposed by the Local Health Authority. Aggregated data may be made available from the corresponding author upon reasonable request and with the permission of the data-providing authority.

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
