# Peer review of "Assessment of Food Hygiene Non-Compliance and Control Measures: A Three-Year Inspection Analysis in a Local Health Authority in Southern Italy"

_foods, 2025, doi:10.3390/foods14193364_

Round 1
Reviewer 1 Report (Previous Reviewer 2)
Comments and Suggestions for Authors
General comment
Thank you for improving the quality of the manuscript.
While including necessary information, it is essential to reduce redundancy. Delete any passages where the same content is repeated. For example, the term “28.6%” is repeated four times in the discussion section. Additionally, the term “Finally” is repeated twice in the discussion section.
1. Material and Methods
Authors should clearly indicate what materials were analyzed and describe how the samples were taken from the target materials.
Since ensuring reproducibility is crucial in research, it is advisable to document details such as which database was used, what SQL queries were employed to create the analysis dataset, and what code was written to analyze the dataset.
2. Figure 1
It goes without saying, it would be better to clearly indicate that the vertical axis represents the number of cases, or it refers to the number of cases confirmed within each year and the horizontal axis represents the year.
3. Conclusion
The conclusion is too lengthy. In the conclusion, briefly state the results of your examination regarding the research hypothesis. Move the discussion to the discussion section.
4. References
Ref. 9
Information on the creating agency is required.
Author Response
We thank you for the insightful feedback and constructive comments on our manuscript.
We have carefully considered all the suggestions and believe that the revisions have significantly improved the quality and clarity of our paper. Below is a point-by-point response to the reviewer's comments, detailing the changes we have made in the revised manuscript.
Comment 1: Thank you for improving the quality of the manuscript. While including necessary information, it is essential to reduce redundancy. Delete any passages where the same content is repeated. For example, the term “28.6%” is repeated four times in the discussion section. Additionally, the term “Finally” is repeated twice in the discussion section.
Response 1: We agree that redundancy should be minimized to improve readability. We have thoroughly revised the Discussion section to remove repetitive phrases and consolidate related points.
Comment 2: Authors should clearly indicate what materials were analyzed and describe how the samples were taken from the target materials. Since ensuring reproducibility is crucial in research, it is advisable to document details such as which database was used, what SQL queries were employed to create the analysis dataset, and what code was written to analyze the dataset.
Response 2: We appreciate the reviewer's focus on methodological rigor and reproducibility. We have revised the Materials and Methods section to provide greater clarity.
Comment 3: It goes without saying, it would be better to clearly indicate that the vertical axis represents the number of cases, or it refers to the number of cases confirmed within each year and the horizontal axis represents the year.
Response 3: We agree that the figure caption can be made more explicit. As suggested, we have revised the caption for Figure 1 to improve clarity.
Comment 4: The conclusion is too lengthy. In the conclusion, briefly state the results of your examination regarding the research hypothesis. Move the discussion to the discussion section.
Response 4: We agree that the original Conclusion section contained elements that were better suited for the Discussion. We have substantially revised and shortened the Conclusion.
Comment 5: Ref. 9. Information on the creating agency is required.
Response 5: we fixed it
Reviewer 2 Report (Previous Reviewer 3)
Comments and Suggestions for Authors
The authors have made very careful revisions and improvements to the manuscript, and addressed all my comments and questions. The manuscript is suitable for publication in Foods now.
Author Response
We are delighted to learn that youour revisions satisfactory and that the manuscript is now considered suitable for publication in Foods.
Thank you for your support
This manuscript is a resubmission of an earlier submission. The following is a list of the peer review reports and author responses from that submission.
Round 1
Reviewer 1 Report
Comments and Suggestions for Authors
I would like to congratulate the authors for the idea of conceiving this study. An in-depth analysis of non-compliance with food hygiene regulations among FBOs is both timely and valuable, as it provides important insights into the types of non-conformities identified by local health authorities—and presumably also by food safety agencies—in Southern Italy.
The paper is interesting but rather clumsily written.
I have written some suggestions as a way to further improve the study. Below are my specific comments.
- I would like to ask: During the inspections, were standardized evaluation/inspection forms used—those officially adopted at the national level in Italy (is it possible to view a sample form)? Or were the inspections based on specific thematic control plans focused on food hygiene?
If not, could you at least provide more detail on which specific aspects of Reg. EC No. 852/2004 were assessed, considering how comprehensive it is?
- Lines 26 and 107 seem to contain a mistake or a gap: “Inspections were carried out in accordance with Reg. EC No. 852/2004, monitoring Critical Control Points (CCPs).” However, CCPs can only be monitored within inspections in a limited way. In practice, CCPs are specific process parameters that are continuously monitored by FBOs themselves, using electronic or visual methods with proper recordkeeping, and are designed to ensure that risks associated with a given technological step remain under control.
For instance, a microbiological CCP parameter (because in the article you talk about food hygiene non-compliance) is very difficult to monitor in real time. That said, you correctly mentioned later that the implementation of GHPs was assessed—this is much more aligned with the actual scope and feasibility of official inspections.
You are absolutely right that the HACCP system represents a fundamental tool for risk management in food production. However, according to EC Regulation No. 852/2004, Article 5, food business operators are required to develop, implement, and maintain one or more procedures based on HACCP principles.
That said, it may be more appropriate to refer primarily to the Guides to Good Hygiene Practice, as recommended in Chapter III of the Regulation, which encourages each Member State to develop and implement such guides.
While HACCP is not mandatory in all cases, basic hygiene procedures and good practices are obligatory. Therefore, it would be more accurate to emphasize compliance with hygiene requirements and national guides, rather than implying that HACCP implementation is universally required.
- Additionally, the non-compliances related to food hygiene regulations could have been categorized as follows: facility structure, transportation, equipment, waste management, personnel, food products, food packaging and packaging materials, HACCP, and other …. Such a classification would have given us a clearer understanding of the source and nature of the identified non-conformities.
- I noticed that the article does not reference one of the fundamental legal texts in the field of food safety: Reg. EC no. 178/2002 which lays down the general principles and requirements of food law, establishes the European Food Safety Authority (EFSA), and sets out procedures in matters of food safety.
Given its foundational role in shaping the EU food safety framework, including risk analysis, traceability, and the responsibilities of food business operators, a brief reference to this regulation would enhance the legal completeness and contextual grounding of the study.
- Since you mention the HACCP system, it would be valuable to provide, at the level of the investigated area in Italy, a summary or categorization such as:
- Number of FBOs with a certified HACCP system
- Number of FBOs with an implemented HACCP system
- Number of FBOs currently in the process of implementing the HACCP system
- Number of FBOs with HACCP documentation developed but not yet implemented
- Number of FBOs that meet preliminary requirements but still need to implement their HACCP system
- Number of FBOs that comply with preliminary requirements
Such a detailed categorization would provide a clearer understanding of the status of HACCP adoption and the nature of non-compliances within the region.
- It would have been valuable to include a reference to Italy’s national legislation that complements EU regulations, particularly in the area of food hygiene and safety. Italian laws and ministerial decrees often provide specific implementation details, enforcement mechanisms, and regional adaptations that are essential for understanding how EU food safety regulations are applied in practice.
Highlighting these national provisions would have offered readers a clearer picture of Italy’s legal framework, as well as the institutional responsibilities and operational procedures followed by Local Health Authorities. Furthermore, it would help contextualize the particular challenges and practices observed in Southern Italy within the broader national regulatory landscape.
- It is ok that the statistical analysis was based on a large sample of 579 enterprises encompassing 1,469 distinct production activities. However, for a clearer understanding of the situation in the investigated area, it would be helpful to include data on the total number of registered FBOs in the region, as found in official databases, along with a categorization of the types of activities (i.e., sectors of activity).
The sentence “The sample included 579 enterprises comprising a total of 1,469 distinct production activities” does not clarify the proportional distribution or weight of specific activity types.
Are the production units large-scale or small-scale? Do they operate in the animal-based or non-animal-based sector?
Moreover, if Reg. EC No. 853/2004 was not cited, this may imply that the sample did not include establishments authorized for intra-Community trade, which would be an important limitation to specify.
- Is the principle of flexibility applied, or are there exemptions (derogations) granted for certain establishments in Southern Italy when specific constraints exist? You mention heterogeneous food production methods, tourism-driven demand, and socio-economic disparities—do these factors justify such exemptions? European Food Safety Authority provides guidance and conducts risk assessments related to food safety, including inspections. They emphasize risk-based approaches to food safety Are the non-compliances related exclusively to microbiological safety criteria for foods of animal origin (Regulation (EC) No. 2073/2005, as amended) and non-animal origin foods? Or do you also address other physicochemical parameters? This is not clearly specified in the paper.
- Please provide detailed information on the product categories (meat, milk, fish, honey, eggs) and the types of microorganisms identified during inspections that detected hygiene non-compliances, such as Listeria monocytogenes, Salmonella spp., Staphylococcal enterotoxin, Cronobacter spp., E. coli, etc.
If so, a much more detailed statistical breakdown of which products showed non-compliances would be necessary.
For non-animal origin products, besides microbiological criteria and sanitation tests, are inspections also addressing other non-compliances such as contaminants, additives, radioactive contamination, treatment with ionizing radiation, genetically modified organisms, etc.?
Clarification on these points would greatly enhance the understanding of the scope and depth of the inspection management findings and promote flexibility in inspections, particularly for small retail businesses.
- Lines 136–169 – Results: The paper reads more like a very general activity report over 4 years. It does not provide precise data or delve into important details that would demonstrate a thorough evaluation of non-compliance in food hygiene (I assume only microbiological criteria were considered). Which microorganisms were detected and in what proportions?
Were there any cases of foodborne illness linked to the hygiene non-compliances identified?
- Lines 170–229 – Discussion: The discussion mainly focuses on interpreting the obtained statistical data. There are no comparisons at the European or global levels, nor with other regions in Italy or other countries. The discussion is poorly written and structured, requiring a complete rephrasing.
- Conclusions: The conclusions should be strictly related to the study conducted and possibly suggest further research directions. I do not understand why references 20 to 30 are cited in the conclusions section of the article.
Author Response
Comment 1: During the inspections, were standardized evaluation/inspection forms used—those officially adopted at the national level in Italy (is it possible to view a sample form)? Or were the inspections based on specific thematic control plans focused on food hygiene?
Response 1: We thank the reviewer for this important clarification. We have amended the manuscript to provide more detail on the inspection procedures. The inspections were conducted by qualified LHA personnel following a standardized institutional checklist.
Comment 2: Lines 26 and 107 seem to contain a mistake or a gap: “Inspections were carried out in accordance with Reg. EC No. 852/2004, monitoring Critical Control Points (CCPs).” However, CCPs can only be monitored within inspections in a limited way. In practice, CCPs are specific process parameters that are continuously monitored by FBOs themselves, using electronic or visual methods with proper recordkeeping, and are designed to ensure that risks associated with a given technological step remain under control.
Response 2: we agree and correct terminology to accurately reflect the role of the inspector.
Comment 3: Additionally, the non-compliances related to food hygiene regulations could have been categorized as follows: facility structure, transportation, equipment, waste management, personnel, food products, food packaging and packaging materials, HACCP, and other …. Such a classification would have given us a clearer understanding of the source and nature of the identified non-conformities.
Response 3: We agree with the reviewer's point on the principle of flexibility. We have revised the introduction to better reflect this. After the sentence on HACCP, we can add: "It is important to note that Regulation (EC) No. 852/2004 requires food business operators to implement and maintain procedures based on the HACCP principles. However, it also allows for flexibility, particularly for small businesses. In many cases, adherence to officially recognized Guides to Good Hygiene Practice, which are encouraged by the Regulation, is considered sufficient to meet the legal requirements for managing food safety risks, without the need for a formal, complex HACCP system."
Comment 4: I noticed that the article does not reference one of the fundamental legal texts in the field of food safety: Reg. EC no. 178/2002 which lays down the general principles and requirements of food law, establishes the European Food Safety Authority (EFSA), and sets out procedures in matters of food safety.
Response 4: A detailed categorization of non-compliances would indeed provide deeper insights. Unfortunately, the raw data extracted from the LHA's official database for the purpose of this study was aggregated by the type of official action taken by the inspector (e.g., 'long-term prescription,' 'violation report') rather than by the specific nature of the non-compliance (e.g., 'structural,' 'equipment'). Retrospective re-categorization was not feasible. To address this, we have acknowledged this as a limitation of our study in the Discussion section
Comment 5:
- Since you mention the HACCP system, it would be valuable to provide, at the level of the investigated area in Italy, a summary or categorization such as:
- Number of FBOs with a certified HACCP system
- Number of FBOs with an implemented HACCP system
- Number of FBOs currently in the process of implementing the HACCP system
- Number of FBOs with HACCP documentation developed but not yet implemented
- Number of FBOs that meet preliminary requirements but still need to implement their HACCP system
- Number of FBOs that comply with preliminary requirements
Such a detailed categorization would provide a clearer understanding of the status of HACCP adoption and the nature of non-compliances within the region.
Response 5: We have added a reference to Regulation (EC) No. 178/2002.
Comment 6: It would have been valuable to include a reference to Italy’s national legislation that complements EU regulations, particularly in the area of food hygiene and safety. Italian laws and ministerial decrees often provide specific implementation details, enforcement mechanisms, and regional adaptations that are essential for understanding how EU food safety regulations are applied in practice.
Response 6: Regarding the status of HACCP implementation, this level of detail was not available in the aggregated dataset. We have included this as a suggestion for future studies in the Conclusion.
Regarding national legislation, we agree that this context is important.
Comment 7: It is ok that the statistical analysis was based on a large sample of 579 enterprises encompassing 1,469 distinct production activities. However, for a clearer understanding of the situation in the investigated area, it would be helpful to include data on the total number of registered FBOs in the region, as found in official databases, along with a categorization of the types of activities (i.e., sectors of activity).
Response 7: We have expanded the description of our sample to provide better context
Comment 8-10:
- Is the principle of flexibility applied, or are there exemptions (derogations) granted for certain establishments in Southern Italy when specific constraints exist? You mention heterogeneous food production methods, tourism-driven demand, and socio-economic disparities—do these factors justify such exemptions? European Food Safety Authority provides guidance and conducts risk assessments related to food safety, including inspections. They emphasize risk-based approaches to food safety Are the non-compliances related exclusively to microbiological safety criteria for foods of animal origin (Regulation (EC) No. 2073/2005, as amended) and non-animal origin foods? Or do you also address other physicochemical parameters? This is not clearly specified in the paper.
- Please provide detailed information on the product categories (meat, milk, fish, honey, eggs) and the types of microorganisms identified during inspections that detected hygiene non-compliances, such as Listeria monocytogenes, Salmonella spp., Staphylococcal enterotoxin, Cronobacter spp., E. coli, etc.
If so, a much more detailed statistical breakdown of which products showed non-compliances would be necessary.
For non-animal origin products, besides microbiological criteria and sanitation tests, are inspections also addressing other non-compliances such as contaminants, additives, radioactive contamination, treatment with ionizing radiation, genetically modified organisms, etc.?
Clarification on these points would greatly enhance the understanding of the scope and depth of the inspection management findings and promote flexibility in inspections, particularly for small retail businesses.
- Lines 136–169 – Results: The paper reads more like a very general activity report over 4 years. It does not provide precise data or delve into important details that would demonstrate a thorough evaluation of non-compliance in food hygiene (I assume only microbiological criteria were considered). Which microorganisms were detected and in what proportions?
Response 8: we state the scope of the study, which concerns hygiene inspections and not laboratory results.
Reviewer 2 Report
Comments and Suggestions for Authors
- General comments
This study investigated the status of food hygiene inspections conducted by administrative agencies in a region of Italy. The study aimed to promote future improvements by clarifying the actual situation. However, it is unclear whether the results adequately reflect the region's distinctive characteristics.
- Ethical concerns
Although it is stated that the research was conducted with the approval of the administrative agency, an explanation of the procedures required to use the agency's information for research purposes would promote research in other regions. Additionally, if ethical considerations were deemed unnecessary for this research, this should be explicitly stated.
- Introduction
Line 80: “Southern Italy exemplifies a region confronted with unique difficulties …” and line 84 “Contributory factors may include …”
Describing how the unique difficulties identified in this region are related to noncompliance would improve the quality of research hypothesis analyses.
- Material and Methods
Authors should clearly indicate what materials were analyzed and describe how the samples were taken from the target materials.
- Results
- Figure 1
It would be better to clearly indicate that the vertical axis represents the number of cases, or it refers to the number of cases confirmed within each year and the horizontal axis represents the year.
- Figure 2
It would be better to clearly indicate that this is a percentage. Are there any duplicates?
- Discussion
- The study area was selected as a research target because of its distinctive characteristics. Therefore, it would be beneficial to discuss the unique results obtained from the survey in this area and compare them with those from other studies.
- Line 213: “if inclusion criteria were not consistently applied or certain business categories underrepresented.”
Is this assumption correct? In any case, I think it would be better to discuss bias based on quantitative analysis.
- Conclusions
Is this assumption correct? In any case, I think it would be better to discuss bias based on quantitative analysis.
- Data Availability Statement
The statement is missing.
- Acknowledgments
Is there anyone you would like to thank for making this research possible?
- References
- It is necessary to standardize the notation of volumes and issues.
- Ref. 11
- Information on the creating agency is required.
Author Response
We thank the reviewer for his time and valuable comments that will help us improve the clarity and impact of our manuscript.
All of your suggestions are marked in green
Comment 1: General Comments
Response1: We thank the reviewer for this insightful suggestion. We have revised the introduction to more explicitly link the region's unique characteristics to potential non-compliance issues, thereby strengthening our research hypothesis.
Comment 2: Ethical concerns
Response 2: We agree that clarifying the ethical procedure is essential. We have added a new subsection to the manuscript (see 2.5 Ethical Statement)
Comment 3: Introduction
Response 3: We agree that explicitly linking the region's unique characteristics to potential non-compliance patterns strengthens the research hypothesis. We have revised the Introduction to elaborate on how factors such as tourism-driven demand and socioeconomic disparities could directly influence the types of non-compliances observed, thus providing a clearer framework for the analysis of our findings
Comment 4: Materials and Methods
Response 4: We have revised this section to improve clarity and precision.
Comment 5: Results
Response 5: We have revised this section to improve clarity and precision.
Comment 6: Discussion
Response 6: We have added a comparative discussion.
Comment 7: Conclusions
Response 7: missing sections and correct references are now available
Reviewer 3 Report
Comments and Suggestions for Authors
This study focuses on food hygiene non-compliance in a Local Health Authority (LHA) in Southern Italy from 2022 to 2024. It analyzes inspection data from 579 enterprises involving 1,469 production activities to explore trends and challenges in food hygiene supervision. Using descriptive statistics and chi-squared tests, it systematically examines inspection triggers, non-compliance types, and corrective measures based on EC Regulation No. 852/2004. The study reveals regulatory difficulties in Southern Italy caused by production heterogeneity and tourism demand, providing data support for targeted interventions and filling the gap of insufficient systematic analysis in food hygiene supervision in this region. However, this study lacked analysis on the correlation between specific non-compliance types and enterprise characteristics, and insufficient explanation of sample representativeness. The manuscript needs revisions and improvements in the following:
1. The format of reference citations in the text are nonstandard. For example, Line 61, [5][6] should be [5-6]; Line 69, [9][10] should be [9-10].
2. Lines 102-103: It is mentioned that the sample was selected by FHNS to "represent the regional picture in a heterogeneous way", but the specific sampling framework, stratification basis, or sample size determination method are not explained, which may affect the representativeness of the results. It is recommended to supplement details of the sampling logic.
3. Lines 114-120: It mentioned that non-compliances are classified as serious and minor, but all the manuscript does not specify their proportions, differences in corresponding corrective measures, or specific risk differences to public health. It is recommended to add relevant statistical data to enhance the depth of analysis.
4. Lines 145-148: In the description related to Figure 1, there were 463 production activities in 2022 with 130 inspected, so the inspection rate should be approximately 130/463≈28.1%, but the authors state 29.8%. In 2023, there were 570 production activities with 356 inspected, and 356/570≈62.46%, which is consistent with the 62.5% in the article. However, the 2022 data have an obvious deviation and needs to be verified and corrected.
5. Lines 192-200: It was pointed out that long-term prescriptions (28.6%) result from structural issues (such as insufficient infrastructure), but there is no analysis combined with specific cases or data on the distribution differences among different enterprise types (e.g., small and medium-sized vs. large enterprises). It is recommended to supplement classified statistical data to support this conclusion.
In addition, it only describes the proportion of types of corrective measures (such as 28.6% long-term prescriptions and 22.9% violation reports), but does not analyze the actual impact of these measures on subsequent compliance rates). It is recommended to add tracking data to evaluate the long-term effectiveness of the measures and improve the practicality of the research.
6. Lines 201-206: The correlation between multiple production activities (avg. 4.61 production activities per enterprise) and non-compliance is plausible but underexplored. Include regression analysis to quantify this relationship.
7. Lines 231-262: The conclusion section is excessively lengthy and needs to be rewritten and organized. Try to present the most refined and concise summary in one paragraph.
8. The study’s reliance on a single Local Health Authority (LHA) dataset raises concerns about external validity. However, regional variability in inspection protocols, economic conditions, and cultural attitudes toward food hygiene may limit broader applicability. Thus, maybe the authors could consider conducting a comparative analysis with other LHAs.
Author Response
Comment 1: The format of reference citations in the text are nonstandard. For example, Line 61, [5][6] should be [5-6]; Line 69, [9][10] should be [9-10].
Response 1: Format fixed
Comment 2: Lines 102-103
Response 2: We thank the reviewer for highlighting the need for greater clarity on our sampling methodology. We have revised the text to better explain the process. The original wording was imprecise. The study was not based on a statistical sampling method but rather on a census of all official control activities conducted by the FHNS in the specified period. The term 'selected by the FHNS to represent the regional picture' refers to the LHA's own internal strategy for official controls, which aims to ensure heterogeneous coverage across different sectors and geographical areas as part of its institutional mandate. Our dataset therefore includes the complete record of these planned and reactive inspections, reflecting the real-world operational focus of the authority.
Comment 3 : Lines 114-120
Response 3: While the concepts of 'serious' and 'minor' non-compliances guide an inspector's judgment, the aggregated administrative dataset available for our study did not allow for a quantitative breakdown based on this classification.
Comment 4: Lines 145-148
Response 4: We sincerely thank the reviewer for their careful reading and for identifying this calculation error. We have re-examined our source data. The issue was a typo in the total number of production activities for 2022 reported in the draft. The correct total to achieve the 29.8% rate was 436, not 463. We have corrected this in the manuscript.
Comment 5: Lines 192-200
Response 5: two very important points regarding the depth of our analysis are raised
-
Regarding the link between long-term prescriptions and structural issues: Our dataset did not allow for a stratification of non-compliances by enterprise type (e.g., SME vs. large). Therefore, our conclusion is indeed an inference. We have rephrased this in the Discussion to present it as a well-founded hypothesis rather than a proven fact.
- Regarding the long-term effectiveness of measures: "An analysis of the long-term effectiveness of corrective measures would require a longitudinal study design, tracking the recidivism rate of non-compliances over time, which was beyond the scope of our retrospective cross-sectional analysis. We have acknowledged this as a key area for future research in our Conclusion.
Comment 6: Lines 201-206
Response 6:We agree with the reviewer that a regression analysis would be a powerful tool to quantify the relationship between operational complexity and non-compliance rates. However, the objective of our study was to provide a descriptive overview of the LHA's inspection activities and trends. A formal regression analysis would require a different study design and was beyond the scope of this initial investigation.
Comment 7:Lines 231-262: The conclusion section is excessively lengthy and needs to be rewritten and organized. Try to present the most refined and concise summary in one paragraph.
Response 7: fixed
Comment 8:
Response 8: We fully acknowledge the reviewer's point regarding external validity. A major limitation of this study is its reliance on data from a single Local Health Authority. As the reviewer correctly notes, regional variability in inspection protocols, economic conditions, and FBO characteristics is significant across Italy. Therefore, our findings should be interpreted as a detailed case study of a specific context in Southern Italy and may not be directly generalizable to other regions. However, they provide a valuable benchmark and an in-depth look at the challenges that may be faced by other LHAs operating in similar socioeconomic environments. We strongly support the reviewer's suggestion that a future comparative analysis across multiple LHAs is crucial for developing a comprehensive national picture of food safety enforcement
Round 2
Reviewer 1 Report
Comments and Suggestions for Authors
The authors have improved the manuscript by adding certain clarifications and supplementary details; however, I do not perceive a clear novelty or identifiable lessons to be drawn from these findings. The measures undertaken by LHA inspectors are periodically reported to the competent authority and are communicated through quarterly and annual press releases/media.
In summary, the manuscript constitutes a three-year report based on data obtained from the Local Health Authority. Such data, including the number of FBOs and the number of inspections conducted by LHA inspectors, are publicly available in several countries, particularly within the EU.
The distinctive contribution of the manuscript lies in its classification that includes the “reason for inspection.” This section is well-structured and of interest, although its relevance is limited to the local level at which the study was conducted. In practice, the reasons for inspection can be far more complex and depend on the limited competences of the authority in question. For instance, the Rapid Alert System for Food and Feed (RASFF) constitutes only one component of the Alert and Cooperation Network, which manages high-risk cases. Non-compliances posing lower risks are addressed at the European level through the Administrative Assistance and Cooperation Network (AAC). Additionally, for information purposes, another system addressing non-compliances is the EU Agri-Food Fraud Network (FFN).
The authors do not provide detailed clarification for each entry in Tables 1 and 2.
Table 1, presenting the types of non-compliance, is poorly documented (recently added).
The information provided is insufficient. This is the reason why I requested the inclusion of a check-list in the paper, in order to review the standard form used during inspections. However, since it was not prepared by the Authors (but rather by the Central Authority), it could not be included. Had the Authors contributed to the development of an inspection form/check-list, it would have been a strong point of the paper.
There are no comparisons in the discussion section with other regions or countries in the EU or outside the EU.
The paper does not have the necessary complexity to be published, in its current form, in a prestigious journal such as Foods.
Author Response
I appreciate the reviewer's detailed feedback and the opportunity to clarify the manuscript's contributions. We have made revisions to strengthen the paper and address the concerns raised.
On the matter of novelty and lessons learned, our study offers a unique, in-depth analysis of the specific non-compliance issues and the regulatory responses within a Local Health Authority (LHA) in Southern Italy in terms of inspection coverage, long-term prescriptions (28.6%), which suggests the prevalence of "structural or systemic" deficiencies, such as inadequate infrastructure or insufficient staff training. This finding provides a valuable, specific insight into the challenges faced in Southern Italy, which may differ from those in other regions where issues are more often related to documentation. This detailed breakdown is not typically included in public reports and underscores the complexity of risk management at the local level.
We fixed the tables, adding description for each entry in order to make it more clear
However we tried to arrange the manuscript (they are highlighted in red) accordingly to your precious suggestions, hoping that you can agree with the publication of this manuscript, trying to dissuade you from your last statement.
Reviewer 3 Report
Comments and Suggestions for Authors
The manuscript was well revised and improved accordingly.
Author Response
Comment 1: The manuscript was well revised and improved accordingly.
Response 1: Thank you for providing your kind support and suggestions